# A Label-Free Impedimetric Genosensor for the Nucleic Acid Amplification-Free Detection of Extracted RNA of Dengue Virus

**DOI:** 10.3390/s20133728

**Published:** 2020-07-03

**Authors:** Ching-Chou Wu, Hao-Yu Yen, Lu-Ting Lai, Guey-Chuen Perng, Cheng-Rei Lee, Shuenn-Jue Wu

**Affiliations:** 1Department of Bio-industrial Mechatronics Engineering, National Chung Hsing University, No. 145, Xingda Rd., South Dist., Taichung City 402, Taiwan; hugo19810607@hotmail.com (H.-Y.Y.); luting822000@yahoo.com.tw (L.-T.L.); 2Innovation and Development Center of Sustainable Agriculture, National Chung Hsing University, No. 145, Xingda Rd., South Dist., Taichung City 402, Taiwan; 3Department of Microbiology and Immunology, College of Medicine, National Cheng Kung University, No.1, University Rd., Tainan City 701, Taiwan; gperng@mail.ncku.edu.tw; 4Viral & Rickettsial Diseases Department, Naval Medical Research Center, Silver Spring, MD 20910, USA; cheng-rei.r.lee.ctr@mail.mil (C.-R.L.); shuenn-jue.l.wu.civ@mail.mil (S.-J.W.)

**Keywords:** impedimetric genosensor, label-free, probe DNA density, dengue RNA, overhang, nucleic acid amplification-free

## Abstract

Developing rapid and sensitive diagnostic methods for dengue virus (DENV) infection is of prime priority because DENV infection is the most prevalent mosquito-borne viral disease. This work proposes an electrochemical impedance spectroscopy (EIS)-based genosensor for the label-free and nucleic acid amplification-free detection of extracted DENV RNA intended for a sensitive diagnosis of DENV infection. A concentration ratio of 0.04 mM 6-mercaptohexanoic acid (MHA) to 1 mM 6-mercapto-1-hexanol (MCH) was selected to modify thin-film gold electrodes as a link to control the coverage of self-designed probe DNA (pDNA) at a density of 4.5 ± 0.4 × 10^11^ pDNA/cm^2^. The pDNA/MHA/MCH-modified genosensors are proven to improve the hybridization efficiency of a synthetic 160-mer target DNA (160mtDNA) with a 140-mer electrode side overhang as compared to other MHA/MCH ratio-modified genosensors. The MHA(0.04 mM)/MCH(1 mM)-modified genosensors also present good hybridization efficiency with the extracted DENV serotype 1 (DENV1) RNA samples, having the same electrode side overhangs with the 160mtDNA, showing a low detection limit of 20 plaque forming units (PFU)/mL, a linear range of 10^2^–10^5^ PFU/mL and good selectivity for DENV1. The pDNA density-controlled method has great promise to construct sensitive genosensors based on the hybridization of extracted DENV nucleic acids.

## 1. Introduction

The epidemic of dengue, malaria, chikungunya, yellow fever, West Nile virus and Japanese encephalitis, notorious diseases transmitted by mosquitoes [1], have been propelled by climate warming, globalization and increasing international activity [2,3,4]. In particular, dengue virus (DENV) infection is the most prevalent mosquito-borne viral disease in tropical and sub-tropical regions covering 40% of the population of the world in 112 countries [1,5]. Infected patients may exhibit asymptomatic, dengue fever [6,7] or severe dengue, such as dengue hemorrhagic fever and dengue shock syndrome [8]. The clinical presentation of dengue fever is similar to many other common febrile diseases making the correct diagnosis of DENV difficult for attending physicians. Therefore, developing rapid and sensitive diagnostic methods for DENV infection from the onset of febrile symptoms is very important in dengue fever prevention and treatment. 

Presently, many strategies have been developed to detect DENV based on the measurement of specific biomarkers of DENV infection, such as soluble non-structural protein 1 (NS1) [9,10,11], DENV envelope protein [12,13], anti-DENV IgM or IgG antibodies [14] and DENV nucleic acid fragments [15,16,17,18,19,20,21] in patient sera. However, the immunoreactive kits cannot differentiate the serotypes of DENV. Primary infection of patients can be caused by any one of the four serotypes of DENV (DENV1, 2, 3, and 4) and secondary infection of these patients with a heterologous serotype increases the risk of progressing into severe dengue due to antibody-dependent enhancement [22,23] Furthermore, compared with NS1 or anti-DENV IgM, DENV particles are the first to be detected in dengue patient serum post symptom onset [21] The serotype-specific detection of DENV will be more time-effective for DENV diagnosis and more meaningful for clinical treatments.

Reverse transcription polymerase chain reaction (RT-PCR) and RT loop-mediated isothermal amplification have been widely used for nucleic acid amplification and the diagnosis of the DENV serotypes by using specific primers [24] However, RT-PCR performance depends on highly trained personnel, well-controlled temperature, expensive instruments and fluorescence-labeled reagents. In contrast, electrochemical DNA genosensors have attracted wide attention for the detection of nucleic acid hybridization in an easy, rapid and inexpensive way. Anusha et al. reviewed the progress of electrochemical genosensors of dengue, classified in amperommetry, voltammetry and impedimetry [25]. Experimentally, specific sequences used as probe DNA (pDNA) are immobilized on the surface of various electrodes, and then the electrochemical signal, after hybridization, is directly quantified by electrocatalytic electrodes, such as Cu_2_CdSnS_4_/O_2_/Si [17], graphite electrode [18] and indium tin oxide (ITO) [26], and evaluated by using electroactive intercalator, such as methylene blue [19,27], and mediators, such as ferrocyanide/ferricyanide [15,16,20,28]. While most publications on electrochemical DNA genosensors used synthetic DNA (≤200-mer) or nucleic acid amplification RNA fragments (<150-mer) as the target analytes to elucidate the sensing performance of genosensors [15,17,18,19,20,27,28], only one study detected the extracted RNA fragments of DENV [16]. Mills et al. [27] and Lynch III et al. [28] developed four-way junction-based genosensors to solve the influence of overhangs on the detection of synthetic DNA (200-mer) and Zika RNA amplicon (147-mer), respectively. Presently, there is little research exploring the nucleic acid amplification-free detection by a genosensor, because it is difficult to control the total length of directly extracted targets with length-varied overhangs, which may hinder the hybridization efficiency. Jin et al. used the SiO_2_@3-aminopropyltriethoxysilane- functionalized graphene oxide (APTES-GO) modified Pt electrode for pDNA immobilization and measured the increment of interfacial electron transfer resistance (*R*_et_) by using electrochemical impedance spectroscopy (EIS) after hybridizing synthetic target DNA (tDNA) or extracted DENV2 RNA at 60 °C for 5 h [16]. Although the label-free EIS-based genosensor had an ultrasensitive detection limit (1 fM) of RNA, the *R*_et_ signal obtained from the hybridization of 1 fM and 10 pM RNA samples had no significant difference. Moreover, the genosensor did not demonstrate selectivity to different serotypes of DENV. These results show that it is worth the efforts to improve the sensing properties of genosensors for the detection of extracted viral nucleic acids. One of the problems for the genosensors to directly detect the extracted nucleic acids is the difficulty in hybridization of extracted DENV RNA (about 11,000 nucleotides) to the pDNA on a densely pDNA-immobilized genosensor. The excessively long overhangs create the steric and repulsive hindrance on the immobilized pDNA making them inaccessible. Corrigan et al. explored the effect of length-varied solution side overhangs on the hybridization efficiency for the 1.5 μM pDNA(15-mer)-modified genosensors [29]. The EIS signal ratio of hybridizing 30-mer tDNA was larger than that of hybridizing 15, 60 and 100-mer tDNA. Riedel et al. also found that the longer solution side overhangs (10−55-mer) decreased the hybridization efficiency, and the electrode side overhangs (10-mer) also decreased the hybridization [30]. Therefore, developing a genosensor capable of overcoming the steric hindrance of the overhangs in order to detect the extracted DENV RNA is an important challenge for the development of PCR-free dengue diagnostics. In particular, there are only a few studies to explore the effect of pDNA density on the hybridization efficiency of extracted nucleic acids.

This work proposes a surface-modifying method to adjust the coverage of immobilized pDNA. A binary self-assembled monolayer (SAM) with different ratio of 6-mercaptohexanoic acid (MHA) and 6-mercapto-1-hexanol (MCH) was used as a link for the covalent immobilization of NH_2_-terminated pDNA via the activated COOH group of SAM. The relationship between the length of electrode side overhangs and the coverage of pDNA was explored by EIS, and the hybridization efficiency of pDNA coverage-varied genosensors was also discussed. Moreover, the selectivity of genosensors immobilized by DENV1-specific pDNA to different serotypes of DENV was compared.

## 2. Materials and Methods

### 2.1. Reagents and Chemicals

Sodium phosphate dibasic, sodium phosphate monobasic dihydrate, tris(hydroxymethyl)aminomethane (Tris), N-[Tris(hydroxymethyl)methyl]-2-aminoethanesulfonic acid (TES), MHA, MCH, N-(3-dimethylaminopropyl)-N′-ethylcarbodiimide hydrochloride (EDC), N-hydroxysuccinimide (NHS), 2-(N-morpholino) ethanesulfonic acid (MES), potassium chloride (KCl), hexaammineruthenium(III) chloride and bovine serum albumin (BSA) were purchased from Sigma-Aldrich (St. Louis, MO, USA). Potassium hexacyanoferrate(III) (K_3_[Fe(CN)_6_]) and potassium hexacyanoferrate(II) trihydrate (K_4_[Fe(CN)_6_]) 3H_2_O were purchased from Showa Chemical (Tokyo, Japan). All chemicals were of reagent grade and were used without further purification. All solutions were prepared with water purified through a Milli-Q system. 

### 2.2. Design and Production of DNA Samples

A full-length sequence of DENV1 (Hawaii, KM204119) was obtained from GenBank of National Center for Biotechnology Information (NCBI). Compared with DENV2 (16681, KU725663), DENV3 (H87, M93130) and DENV4 (H241, AY947539), a 20-mer specific sequence, found in DENV1 RNA by a CLC Sequence Viewer (Qiagen Bioinformatics (Redwood, CA)), was identified as target DNA (designated as 20mtDNA) with the sequences of 5′-GTACCCTGGTGGTAAGGACT-3′. The 3′ terminus of the 20mtDNA has a 140-mer distance away from the 3′ terminus of full-length DENV1 RNA. The corresponding pDNA is completely complementary to the 20mtDNA with the sequences of 5′-AGTCCTTACCACCAGGGTAC-3′ and is modified with the NH_2_-(CH_2_)_6_ group at the 5′ terminus. In order to realize the effect of electrode side overhang on the hybridization efficiency, a 160-mer target DNA, designated as 160mtDNA, was synthesized according to the last 160-mer sequence from 3′ terminus of DENV1 RNA. The sequence of all tDNA and pDNA is shown in Table 1. All DNA strands were synthesized by Genomics (Taiwan) and purified by high performance liquid chromatography.

### 2.3. Preparation of Genosensor

30-nm Ti and 200-nm Au thin films were patterned as a rectangular electrode on a cleaned glass substrate by lift-off microfabrication techniques. The working area of electrodes was defined as a disk shape with 1.6 mm in diameter by a 7 μm-thick negative photoresist, SU8-3010 (MicroChem Laboratory, Newton, MA, USA). After cleaning the surface of the Au electrodes, the electrodes were dipped in the mixture of MHA and MCH with concentration-varied ratio for 4 h to form a binary SAM. The preparation of genosensor is shown in Scheme 1. The COOH group of MHA was activated by the mixture of 30 mM EDC and 30 mM NHS prepared in 20 mM MES (pH 4.6) for 1 h. Afterwards, a 10 μL aliquot of 4 μM pDNA and 150 mM NaCl-containing 10 mM phosphate buffer (pH 7.0) was dripped on the activated SAM and incubated at 40 °C for 1 h for covalent immobilization. After rinsing with pure water, the pDNA/MHA/MCH/Au electrodes were used as genosensors. Experimentally, 10 μL aliquot of the concentration-varied 20mtDNA or 160mtDNA was dropped on the genosensors for 60 min at room temperature for hybridization.

### 2.4. Measurement of Self-Assembled Monolayer (SAM) Coverage and pDNA Density 

The surface coverage of MHA/MCH binary SAM can be quantified via a one-electron reductive path in an alkaline solution (pH ≥ 11) [31]. The reductive equation is represented as follows:

AuS(CH_2_)_n_X + e^−^→ Au(0) + S(CH_2_)_n_X^−^, where X = COO^−^ and OH.


Hence, the amount of reductive charge arising from Au-S desorption can be counted by integrating the reduction current against time (Q = ∫idt) for the determination of the coverage per cm^2^. The pDNA coverage immobilized at the MHA/MCH-modified electrodes can be calculated by the amount of positively electroactive molecules adsorbed with the negatively charged phosphate group of DNA backbone [32]. In Steel’s study the trivalent cation, ruthenium(III) hexaammine (RuHex), had been proved to quantify pDNA density due to the strong electrostatic association between RuHex cation and DNA phosphate in a low ionic strength (*IS*) electrolyte. The amount of adsorbed RuHex can be measured using chronocoulometry by integrating the reductive current of RuHex. The obtained current is attributed to the diffusion-controlled current, the charging current of the double layer, and the reductive current of adsorbed RuHex. The corresponding charge, Q, is presented in Equation (1).
*Q* = 2*nFAD*^1/2^*C*^*^*t*^1/2^*/π*^1/2^ + *Q*_dl_ + *nFA*Γ_0_(1)

The first term on the right-hand side is obtained from the integrated Cottrell equation with a linear relation with the square root of time (*t*^1/2^). Where *n* (= 1) is the electron-transfer number per RuHex for reduction, *F* is Faraday constant (96485 C/mol), *A* (= 0.02 cm^2^) is the electrode area, *D* is the diffusion coefficient of RuHex, *C^*^* is the bulk concentration of RuHex, *Q*_dl_ is the capacitive charge (C), and Γ_0_ is the surface density (mol/cm^2^) of adsorbed RuHex. The chronocoulometric intercept at *t* = 0 is the sum of *Q*_dl_ and *nFA*Γ_0_. Therefore, the Γ_0_ is determined from the difference in chronocoulometric intercepts at *t* = 0 in the presence and absence of RuHex molecules. The pDNA/MHA/MCH/Au genosensors were respectively immersed in 10 mM Tris buffer (pH 7.4) and 50 μM RuHex-containing 10 mM Tris buffer (pH 7.4) to perform a step potential from 0.1 V to −0.4 V for the calculation of Γ_0_. Afterwards, the pDNA surface density can be calculated according to Equation (2).
Γ_DNA_ = Γ_0_(*z*/*m*)(*N*_A_)(2)
where Γ_DNA_ is the pDNA density (molecules/cm^2^), *m* (=20) is the number of bases in the pDNA, *z* (=3) is the charge of the RuHex molecule, and *N*_A_ is Avogadro’s number. 

### 2.5. Electrochemical Measurements

Cyclic voltammetry (CV), chronocoulometry and EIS were performed with an IM-6 impedance analyzer (Zahner Electrik GmbH, Germany) in a three-electrode system. Pt wire and Ag/AgCl electrode were used as the counter electrode and the reference electrode, respectively. The 1 mM equimolar Fe(CN)_6_^3−/4−^-containing 26 mM TES buffer (pH 7.0) was used to probe the electrochemical properties of the electrode/electrolyte interface. Impedimetric measurement was carried out in the frequency range of 1 Hz to 10 kHz at +0.2 V potential added with a 5 mV amplitude sine wave versus the Ag/AgCl electrode. The IM-6/THALES software package was used for the acquisition of impedance spectra, and the simulation of equivalent circuits. 

### 2.6. Extracted RNA Test 

The commercial E.Z.N.A.^®^ Viral RNA Kit (Omega Bio-tek (Norcross, GA) was used to extract RNA fragments from DENV1 suspension samples ranged from 10^4^ to 10^6^ plaque forming units (PFU)/mL, supplied by the laboratory of the co-author, Dr. Perng. A 150 μL aliquot of the DENV sample was first mixed in 500 μL QVL Lysis Buffer for 7 min, and then slightly centrifugalized. 350 μL aliquot of pure ethanol (99.9%) was added in the mixture, vortexed for 30 s, and then slightly centrifuged. The mixture was transferred to HiBind^®^ RNA Mini Column and centrifuged at 13,000 g for 15 s. This step was repeated until the residual RNA sample was fully transferred to the column. Subsequently, the HiBind^®^ RNA Mini Column was removed to a new collection tube, and 500 μL VHB Buffer was added in the tube for centrifugation at 13,000 g for 15 s. The filtrate was discarded, and the HiBind^®^ RNA Mini Column was removed to a new collection tube. 500 μL Wash Buffer II was added in the tube for centrifugation at 13,000 g for 15 s, and then the filtrate was discarded. This washing step was performed twice. After moving the HiBind^®^ RNA Mini Column to a new tube, centrifugation was performed at 13,000 g for 2 min to remove any ethanol. Finally, the HiBind^®^ RNA Mini Column was removed to a new tube, and a 20 μL aliquot of pure water was added to the center of the HiBind^®^ RNA Mini Column. After performing centrifugation at 13,000 g for 1 min, RNA fragments were collected. The extracted DENV1 RNA was resuspended in 20 μL pure water. After centrifugation of the resuspension, a 130 μL aliquot of 150 mM NaCl-containing 10 mM PBS buffer (pH 6.0) was added to restore the 150 μL initial volume of the DENV sample. Then, 10 μL aliquot of the RNA sample was dropped on the genosensors for 20 min at room temperature for hybridization.

## 3. Results and Discussion

### 3.1. Effect of Binary SAM on pDNA Coverage

Most studies directly immobilize thiolated pDNA on gold electrodes via the formation of an Au-S bond for the preparation of genosensors [29,30,32,33,34]. However, the bases of pDNA have strong non-specific adsorption on the Au surface. MCH of millimolar-scale concentration is commonly used to displace most non-specifically adsorbed pDNA, but some still remained [33]. In this study, the modification of MHA/MCH binary SAM was used as a link for pDNA immobilization to prevent the non-specific adsorption effectively. Figure 1 shows the cyclic voltammograms obtained at the ratio-varied MHA/MCH-modified electrodes. Two cathodic peaks were found in all reductive curves and represented as the desorption behavior of MHA/MCH SAM on the polycrystalline Au substrate. The adsorbing stability of Au-S bond substantially depends on the crystallographic orientation in the order of Au(111) < Au(100) < Au(110) [35]. Therefore, the two reductive peaks at about −0.8 V and −1.1 V were, respectively, attributed to the Au-S desorption in Au (111) face and Au (100) or Au(110) faces. Compared with other MHA/MCH ratios, the first reductive peak obtained at the MHA(0):MCH(1 mM)-modified electrodes was at more negative potential (−0.95 V), resulting from the interaction force between homogeneous MCH molecules stronger than that between MHA and MCH. The coverage of MHA/MCH SAM of 0:1, 0.02:1, 0.04:1 and 0.1:1 ratios calculated by the desorption charge (Q) was, respectively, 9.4 ± 0.3, 9.4 ± 0.1, 10.0 ± 0.2 and 10.2 ± 0.3 × 10^−11^ mol/cm^2^ (*n* = 4). The result showed that the SAM coverage was positively proportional to the total concentration of MHA/MCH mixture. 

Furthermore, EIS was used to estimate the MHA ratio in the binary SAM due to the repulsive force between the negatively charged COOH group of MHA and the Fe(CN)_6_^3−/4−^ [31]. Figure 2a shows the Nyquist plots obtained at bare, 1 mM MCH, 1 mM MHA and ratio-varied MHA/MCH modified electrodes. The impedance spectra measured at the MCH-modified electrode (curve ii of Figure 2a) exhibited a significant linear region, implying the diffusion-controlled behavior of the Fe(CN)_6_^3−/4−^ mediator at the lower frequencies, and a semicircle region, implying the kinetics-controlled behavior [31]. The radius of the semicircle is related to the *R*_et_ of mediator in the electrode/electrolyte interface. The radius of the semicircle obtained at the binary SAM-modified electrodes increased with the increasing MHA ratio, attributed to the electrostatically repulsive force of COOH group of MHA molecules [31]. Furthermore, the semicircle radius of the impedimetric plot of the 1 mM MCH-modified electrode was much smaller than the bare electrode, implying that the hydrophilic OH group of MCH can significantly reduce the *R*_et_. It is worth noting that the impedimetric plots only exist in the semicircle part when the ratio of MHA to MCH is larger than 0.04:1, implying that the higher density of MHA makes the concentration gradient of the mediator harder to produce a difference between the bulk solution and the electrode surface.

Two kinds of equivalent circuits, a modified Randles model (inset of Figure 2b) and a 1R//C model (inset of Figure 2c), were used to simulate the impedimetric results with and without the diffusion-controlled region, respectively. The equivalent circuit of modified Randles model consists of four elements: the solution resistance (*R_s_*), the Warburg impedance (*Z_w_*), the constant phase element (*CPE*) of electrical double layer and *R*_et_. Because the double layer is not an ideal capacitor (*C_dl_*) due to the Faradic current leakage of Fe(CN)_6_^3−/4−^ mediator, *CPE* was used to replace *C_dl_* in the equivalent circuit, especially for the surface-modified electrodes [34]. The impedance of the *CPE* can be presented as *Z_CPE_(ω) = Z_0_(jω)^−α^*, where *Z_0_* is a constant, *j* is an imaginary number, *ω* is the angular frequency, and 0 < *α* < 1. When *α* is closer to 1, the *CPE* becomes more capacitive. When the impedimetric spectrum only contains the kinetics-controlled part, the 1R//C model is used to explain the EIS behavior, such as MHA(0.1 mM)/MCH(1 mM)-modified electrodes, which consist of one resistor (*R_s_*) in series with one parallel circuit comprising a resistor (*R*_et_) and a capacitor (*CPE*). Figure 2b,c, respectively, shows the Bode plots of experimental data measured at the MHA(0.04 mM)/MCH(1 mM)/Au electrode and the MHA(0.1 mM)/MCH(1 mM)Au electrode, and the corresponding computer fitting using the modified Randles and the 1R//C equivalent circuit. The fitting results show good consistency with the experimental measurement with a mean error of less than 0.2% and a maximum error of 2.2% for all fitting data. After the fitting of equivalent circuits, the *R*_et_ value obtained at bare, MCH(1 mM), MHA(0.02 mM)/MCH(1 mM), MHA(0.04 mM)/MCH(1 mM), MHA(0.1 mM)/MCH(1 mM) and MHA(1 mM) modified electrodes was, respectively, 13 ± 0 kΩ, 1 ± 0 kΩ, 2 ± 0 kΩ, 10 ± 0 kΩ, 156 ± 1 kΩ and 5578 ± 164 kΩ (*n* = 3). The smallest *R*_et_ value obtained at the 1 mM MCH-modified electrodes is attributed to the good hydrophilicity of MCH hydroxyl group. Moreover, the *R*_et_ value actually increased with the increasing ratio of MHA in the binary SAMs due to the repulsively electrostatic force from the COO^−^ group of MHA.

The MHA ratio of SAM could determine the density of immobilized pDNA. The coulometric method was used to measure the pDNA density via the reductive charge of RuHex adsorbed on the phosphate group of pDNA, after immobilization of pDNA on the MHA-activated surface. Figure 3 shows the chronocoulometric curves measured from the different pDNA/MHA/MCH/Au electrodes in the absence and presence of RuHex. Here, the denotation of MCH concentration is omitted for a concise reading because the concentration, 1 mM, of MCH used for all MHA/MCH modification was the same. The fitting range for the 1 mM MHA SAM was from 0.2 s^1/2^ to 0.7 s^1/2^, and for the all MHA/MCH binary SAM was from 0.1 s^1/2^ to 0.4 s^1/2^. The intercepts of fitting line at *t* = 0 obtained in the RuHex-containing buffer and the Tris buffer, respectively, were used for the calculation of Γ_0_ [32]. The Γ_0_ measured on the pDNA/MHA(1 mM)/Au, the pDNA/MHA(0.1 mM)/MCH/Au, pDNA/MHA(0.04 mM)/MCH/Au and the pDNA/MHA(0.02 mM)/MCH/Au electrodes was, respectively, 175.8 ± 3.4, 16.1 ± 6.1, 4.5 ± 0.4 and 3.2 ± 0.5 × 10^11^ pDNA/cm^2^ (*n* = 3) and the interval between two adjacent pDNA molecules on the electrodes was respectively calculated as 2.4 nm, 7.9 nm, 14.9 nm and 17.7 nm by assuming a homogeneous distribution of pDNA on the SAM surface. The result shows that the Γ_0_ of immobilized pDNA is positively correlated to the MHA ratio of MHA/MCH SAMs. Nevertheless, the 2.4 nm wide interval between two adjacent pDNA molecules on the pDNA/MHA(1 mM)/Au genosensors is adverse to the tDNA hybridization because the diameter of double-stranded DNA is 2 nm. Moreover, the electrostatically repulsive force of the phosphate group in the hybridized DNA backbone may hinder the hybridization of another suspended tDNA. The thickness (=3.04/(*IS*^0.5) A°) of the electric double layer is calculated as 0.7 nm with 190 mM *IS* in the 150 mM NaCl-containing 10 mM phosphate buffer (pH 7.0) solution. Steel et al. also demonstrated that the pDNA density greater than 4 × 10^12^ pDNA/cm^2^ significantly reduced the hybridizing efficiency of complementary tDNA [32]. Therefore, the modification of MHA (1 mM) is not used to construct the genosensors.

### 3.2. Effect of pDNA Density on Hybridization of Length-Varied tDNA

The interval between the immobilized pDNA molecules could affect the hybridization degree [32]. Figure 4 and Figure 5 show the impedimetric spectra obtained at the genosensors modified with the MHA/MCH SAM of different ratios for the hybridization of 20mtDNA and 160mtDNA, respectively. The results presented that the radius of all semicircle increased with the tDNA hybridization. After simulation of the 1R//C model for the genosensors modified by MHA(0.1 mM)/MCH and MHA(0.04 mM)/MCH, and the modified Randles model for the genosensors modified by MHA(0.02 mM)/MCH, the *R*_et_ increment (Δ*R*_et_ = *R*_et-Tdna_ − *R*_et-pDNA_) and the Δ*R*_et_ ratio (%)(= Δ*R*_et_/*R*_et-pDNA_ × 100) of genosensors after the hybridization of 1 pM and 1 nM tDNA are compared in Table 2. The Δ*R*_et_ obtained from the three kinds of MHA/MCH genosensors increased with the tDNA concentration, and the Δ*R*_et_ obtained at the denser pDNA-immobilized-genosensors increased more significantly. Moreover, the Δ*R*_et_ obtained at the 160mtDNA-hybridized genosensors was much larger than that obtained at the 20mtDNA-hybridized genosensors, resulting from the steric and repulsive hindrance of the 140-mer electrode side overhang of 160mtDNA from the permeation of Fe(CN)_6_^3−/4−^ to the electrode. Riedel et al. found a similar phenomenon when using a 10-mer electrode side overhang tDNA to obtain the larger Δ*R*_et_ ratio than a 5-mer electrode side overhang [30]. 

Compared to the Δ*R*_et_, Δ*R*_et_ ratio is more often used to compare the Δ*R*_et_ increment between different modified electrodes by normalizing the initial impedimetric status of pre-hybridized genosensors [29,30]. It is worth noting that the Δ*R*_et_ ratio obtained at the MHA(0.04 mM)/MCH-modified genosensors was larger than that obtained at the MHA(0.1 mM)/MCH- and MHA(0.02 mM)/MCH-modified genosensors. Moreover, the Δ*R*_et_ ratio of 160mtDNA hybridization was much larger than that of 20mtDNA hybridization. The results indicated that the pDNA density of MHA(0.04 mM)/MCH-modified genosensors was more effective for the hybridization of 160mtDNA than other MHA/MCH-modified genosensors. The smaller interval between the pDNA molecules of the MHA(0.1 mM)/MCH-modified genosensors was adverse to the hybridization of long electrode side overhangs of 160mtDNA. Although the larger interval between the pDNA molecules of the MHA(0.02 mM)/MCH-modified genosensors was beneficial to the hybridization of 160mtDNA, the reduced pDNA density produced lower Δ*R*_et_ ratio than that obtained at the other MHA/MCH-modified genosensors and was easier to reach saturated hybridization. The increment of Δ*R*_et_ ratio from 1 pM to 1 nM 160mtDNA measured at the MHA(0.02 mM)/MCH-modified genosensors was only 4.9%, which was smaller than that (20.2%) measured at the MHA(0.04 mM)/MCH-modified genosensors. The results suggested that the MHA(0.04 mM)/MCH-modified genosensors was more suitable for the hybridization of extracted DENV1 RNA due to the similar electrode side overhangs with the 160mtDNA. 

### 3.3. Sensing Properties of Genosensors for Extracted RNA

The MHA(0.04 mM)/MCH-modified genosensors were used to measure the extracted RNA samples. Figure 6a shows the Nyquist plots obtained at the pDNA/MHA(0.04 mM)/MCH-modified genosensors followed by time-lapse hybridization of RNA samples extracted from 10^4^ PFU/mL DENV1. The increasing radius of semicircle suggested the increase of *R*_et_ due to the extracted RNA hybridization. Figure 6b shows the time-lapse change of Δ*R*_et_ ratio (*n* = 3) corresponding to the fitting result of Figure 6a. The Δ*R*_et_ ratio increased with time. The Δ*R*_et_ ratio (142.1 ± 3.1%) at 20 min had smaller relative standard deviation (RSD) of 2.2%, implying better reproducibility, and the increment of Δ*R*_et_ ratio slowed down after 20 min hybridization. Therefore, 20 min hybridization for each concentration was used to test the calibration curve. Furthermore, the Δ*R*_et_ ratio (142.1%) of 20 min hybridization, compared to the *R*_et_ value (8.7 ± 0.8 kΩ) of pre-hybridized genosensors with the background noise of 8.7% RSD, was significantly larger than the background noise.

Figure 7a presents the Nyquist plots for the genosensors before and after hybridizing the RNA samples extracted from the concentration-varied DENV1. After fitting, Figure 7b shows the corresponding calibration curve obtained from three individual genosensors. The result presented that the genosensors had a good linearity (*R* = 0.9960) in the range of 10^2^–10^5^ PFU/mL with a regressive equation of Δ*R*_et_ ratio = 33.3 log[DENV]−43.1 and saturated at the 10^5^ PFU/mL. The *R*_et_ value of pre-hybridized genosensors was 8.5 ± 0.2 kΩ, implying the calculated limit of detection (LOD) was about 20 PFU/mL (S/N > 3). Generally, 1 PFU of DENV is equivalent to 600–1000 copies of virus particles in Dr. Perng’s laboratory, which depends on the virus strains, culture conditions and the stored time of the virus [36,37]. The 20 PFU/mL LOD is equal to 20–33 aM RNA copies, which is much lower than that (1 fM extracted RNA of DENV2) of the EIS-based pDNA/SiO_2_@APTES-GO/Pt genosensors [16]. The low LOD was attributed to the very long solution side overhangs and electrode side overhangs of hybridized RNA to hinder the permeation of mediator to the electrode surface, significantly increasing *R*_et_. The Δ*R*_et_ ratio of 10^4^ PFU/mL obtained in the calibration curve was different from that presented in the time-lapse hybridization, partly attributed to the different amount of complementary fragments of extracted RNA and the different hybridizing procedures. As expected, it was difficult to obtain the same copy number of complementary fragments in each extracting procedure. Furthermore, in the previous study [16] the genosenors required a 5 h hybridization incubation and the EIS signal still could not distinguish extracted DENV2 RNA concentrations between 1 fM and 10 pM after hybridization. In contrast, the pDNA/MCH(0.04 mM)/MCH-modified genosensors directly detected the extracted RNA of lower concentration (20 aM) in shorter hybridization time (20 min). According to the reviews by Anusha et al. [25] and Darwish et al. [38], while the electrochemical immunoassays for the detection of DENV NS1, envelope protein and particles can have extremely low LOD down to less than 1 PFU/mL, it remains difficult to distinguish the serotypes of DENV.

The selectivity test of genosensors is shown in Figure 8. The hybridization efficiency of the extracted RNA samples of each DENV serotype were respectively measured by three individual genosensors. The Δ*R*_et_ ratio (56.4 ± 7.9%) for DENV1 is significantly larger than that for DENV2 (11.4 ± 3.4%), DENV3 (17.4 ± 2.3%) and DENV4 (14.7 ± 2.9%) (*p* < 0.05 by student *t*-test). Moreover, the Δ*R*_et_ ratio value for the mixture of DENV1+2, DENV1+3 and DENV1+4 was, respectively, 58.4 ± 4.4%, 60.3 ± 5.0% and 57.8 ± 2.0%. The calculated selectivity (= DENV1/DENV mixture) was in the range of 93.5% to 97.6%, indicating the good selectivity of the self-designed pDNA-immobilized genosensors to DENV1. Furthermore, real-time PCR method was used to identify the specificity of self-designed pDNA as shown in Figure 9. The performance of real-time PCR is according to the handbook of KAPA SYBR^®^ FAST qPCR Master Mix (2X) kit. The self-designed pDNA was used as a specific backward primer, and the sequence of 5′-GGTTAGAGGAGACCCCTCCC-3′ was used as a universal forward primer, which is suitable for all DENVs. The result shows that the threshold cycle for DENV1 is much smaller than that for DENV2, 3 and 4, implying that the self-designed pDNA is very specific to DENV1. Moreover, Table 3 compares this study with previous electrochemical genosensors for the DENV detection. Most DENV genosensors detected synthesize DNA [15,18,19,20,39] fragments and nucleic acid amplicons [26,39,40,41]. Although Chen et al. could obtain as low as 2 PFU/mL LOD, the nucleic acid amplification was necessary [40]. Except for Jin’s study [16] and this study, there are few studies to directly detect the extracted RNA without the process of nucleic acid amplification. Moreover, the pDNA/MHA(0.04 mM)/MCH(1 mM)/Au genosensors have better sensing properties than Jin’s sensor.

## 4. Conclusions

The ratio-varied MHA/MCH binary SAM can be used as a link to control the density of immobilized pDNA. In particular, the pDNA/MHA(0.04 mM)/MCH-modified genosensors improved the hybridization efficiency for 160 mtDNA having 140-mer electrode side overhangs via the EIS analysis. The genosensors designed in this study could detect extracted DENV1 RNA after 20 min hybridization with a low LOD of 20 PFU/mL, good detection linearity from 10^2^ to 10^5^ PFU/mL and good selectivity for DENV1. This study demonstrates that the combination of well-controlled pDNA density and the self-designed pDNA sequence capable of reducing the electrode side overhang length of extracted RNA makes the detection of extracted DENV1 RNA feasible. As expected, this method has great promise to construct label-free EIS-based genosensors for the optimal hybridization of extracted nucleic acids without the process of nucleic acid amplification.

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
