# Peer review of "A Label-Free Impedimetric Genosensor for the Nucleic Acid Amplification-Free Detection of Extracted RNA of Dengue Virus"

_sensors, 2020, doi:10.3390/s20133728_

Round 1

Reviewer 1 Report

The paper by Wu et al. “A Label-Free Impedimetric Genosensor for The Polymerase Chain Reaction-Free Detection of Extracted RNA of Dengue Virus” reports on a new approach to modifying a thin-film electrode for a controlled coverage with a DNA probe. This is reported to improve the hybridization efficiency for long DNA targets. The developed approach of electrode’s modification allowed to detect a long genomic RNA extracted from dengue virus particles.

Comments:

  1. Please make sure all the figures and figure legends are not shifted and are clearly visible.
  2. The authors report the linear dynamic range to be 10^2-10^5 PFU/mL, while the calculated LOD is 20 PFU/mL, which is outside the linear dynamic range. If the calculated LOD value is indeed lower than the linear dynamic range, it does not correctly represent the LOD. Please double check and correct if needed.
  3. In the Introduction section, the authors mentioned PCR techniques and list LAMP as an example of such techniques, but this is not correct. LAMP is a polymerization-based target-amplification method, but it is carried out at one and the same temperature, unlike PCR; relies on 4-6 pairs of primers, unlike PCR; the amplicons are a mixture of dsDNA fragments, unlike PCR.
  4. In the same paragraph of Introduction, the authors use the terms “primer” and “probe” as synonyms, which is not correct. A probe hybridizes to a complementary sequence of the target and reports the target’s present, while a primer hybridizes to a complementary sequence of the template and gets elongated due to a DNA polymerase-catalyzed nucleotide polymerization.
  5. The authors should cite and discuss the following papers, where electrochemical detection of longer than 35-nt targets has been reported: Biosensors and Bioelectronics 2018, 109, 35-42; Analytical Chemistry 2019, 91 (21), 13458-13464.
  6. Please make sure the abbreviation is explained when it is first introduced. For example, on page 2, pDNA is mentioned, but it is not explained what this means.
  7. Please double check and correct, if needed, that an SH-group is required at the 5’-end of the 20-mer target DNA (page 3, section 2.2).
  8. In the equation mentioned in Section 2.4, the carboxyl group is shown as protonated, which would be minimal at pH 11.
  9. In the legend to Figure 5, the sensor is referred to as “aptasensor”. Is it based on the aptamer sequence?
  10. When the authors studied the time-dependence of the signal triggered by RNA extracted from 10^4 PFU/mL, the best reproducibility was observed for 20 min, and this time point was selected for subsequent experiments. Why it was only for 20 min that the data was so consistent? Why not for other time points?
  11. Could you please cite the source of a claim that “Generally, 1 PFU of DENV is equivalent to 600−1000 copies of virus particles”? I believed that 1 PFU corresponds to 1 viral particle, like 1 CFU corresponds to 1 bacterial cell, for example. Is it specific for DENV or for any virus?
  12. Please explain how deltaRret ratio was calculated. What was the value for no-target control? If not zero, present the data for no-target control in Figure 8. Without listing the no-target control signal, it cannot be claimed “superior specificity”. Superior specificity means that all other serotypes would trigger the signal at the no-target level.
  13. How in practice to interpret the results of the sensor? For example, if the signal is ~20%, how to differentiate between 10^2 PFU/mL DENV1 and 10^4 PFU/mL of DENV3?
  14. How the sensor would respond to a sample with a mixture of serotypes (e.g. DENV1 and DENV4, or DENV2 and DENV3)?

Author Response

Comments and Suggestions for Authors

The paper by Wu et al. “A Label-Free Impedimetric Genosensor for The Polymerase Chain Reaction-Free Detection of Extracted RNA of Dengue Virus” reports on a new approach to modifying a thin-film electrode for a controlled coverage with a DNA probe. This is reported to improve the hybridization efficiency for long DNA targets. The developed approach of electrode’s modification allowed to detect a long genomic RNA extracted from dengue virus particles.

Comments:

  1. Please make sure all the figures and figure legends are not shifted and are clearly visible.

Response: Thank you for the reminder. All the figures and legends have been adequately adjusted.

  1. The authors report the linear dynamic range to be 10^2-10^5 PFU/mL, while the calculated LOD is 20 PFU/mL, which is outside the linear dynamic range. If the calculated LOD value is indeed lower than the linear dynamic range, it does not correctly represent the LOD. Please double check and correct if needed.

Response: The calculated LOD, defined as the triple of standard deviation of the blank measurement according to IUPAC definition, is obtained from the regressive equation of dynamic range. Figure 7 shows the Nyquist plots of EIS measurement and ΔRet values obtained from the blank and the 10^2-10^6 PFU/mL DENV1 samples. Therefore, the whole detecting range is from 0 to 10^6 PFU/mL. However, the linear range is only in the range of 10^2-10^5 PFU/mL. Then, the LOD was calculated by the regressive equation according to the IUPAC definition. Furthermore, in other sensor-related articles [18, 20] the calculated LOD may be lower than the dynamic range. The LOD calculation is correct.

[18] N. Oliveira, E. Souza, D. Ferreira, D. Zanforlin, W. Bezerra, M.A. Borba, M. Arruda, K. Lopes, G. Nascimento, D. Martins, M. Cordeiro, J. Lima-Filho, A sensitive and selective label-Free electrochemical DNA biosensor for the detection of specific dengue virus serotype 3 sequences, Sensors (Basel) 15 (2015) 15562–15577.

The linear range was between 10 nM and 100 nM, and the calculated LOD was 3.09 nm.

[20] S. Tripathy, S.R. Krishna Vanjari, V. Singh, S. Swaminathan, S.G. Singh, Electrospun manganese (III) oxide nanofiber based electrochemical DNA-nanobiosensor for zeptomolar detection of dengue consensus primer, Biosens. Bioelectron. 90 (2017) 378–387.

The dynamic detection range was between 1 aM and 1 mM, and the calculated LOD was 0.12 aM.

  1. In the Introduction section, the authors mentioned PCR techniques and list LAMP as an example of such techniques, but this is not correct. LAMP is a polymerization- based target-amplification method, but it is carried out at one and the same temperature, unlike PCR; relies on 4-6 pairs of primers, unlike PCR; the amplicons are a mixture of dsDNA fragments, unlike PCR.

Response: Thanks for the suggestion. On page 2 the sentence has been revised as “Reverse transcription polymerase chain reaction (RT-PCR) and RT loop-mediated isothermal amplification have been widely used for nucleic acid amplification and the diagnosis of the DENV serotypes by using specific primers [24]”.

  1. In the same paragraph of Introduction, the authors use the terms “primer” and “probe” as synonyms, which is not correct. A probe hybridizes to a complementary sequence of the target and reports the target’s present, while a primer hybridizes to a complementary sequence of the template and gets elongated due to a DNA polymerase-catalyzed nucleotide polymerization.

Response: Thanks for the suggestion. On page 2 the sentence has been revised as “Experimentally, specific sequences used as probe DNA (pDNA) are immobilized on the surface of various electrodes,” on page 2.

  1. The authors should cite and discuss the following papers, where electrochemical detection of longer than 35-nt targets has been reported: Biosensors and Bioelectronics 2018, 109, 35-42; Analytical Chemistry 2019, 91 (21), 13458-13464.

Response: The suggested articles, “Analytical Chemistry 2019, 91 (21), 13458-13464” and “Biosensors and Bioelectronics 2018, 109, 35-42” are added in reference list as Ref.27 and Ref.28 respectively. The related description is showed on page 2 as below, “Experimentally, specific sequences used as probe DNA (pDNA) are immobilized on the surface of various electrodes, and then the electrochemical signal, after hybridization, is directly quantified by electrocatalytic electrodes, such as Cu2CdSnS4/O2/Si [17], graphite electrode [18] and ITO [26], and evaluated by using electroactive intercalator, such as methylene blue [19,27], and mediators, such as ferrocyanide/ferricyanide [15,16,20,28]. While most publications on electrochemical DNA genosensors used synthetic DNA (£200 nt) or nucleic acid amplification RNA fragments (<150 mer) as the target analytes to elucidate the sensing performance of genosensors [15,17-20,27,28], only one study detected the extracted RNA fragments of DENV [16]. Mills et al. [27] and Lynch III et al. [28] developed four-way junction-based genosensors to solve the influence of overhangs on the detection of synthetic DNA (200 mer) and Zika RNA amplicon (147 mer) respectively. Presently, there are few researches to explore the nucleic acid amplification-free detection by a genosensor, because the total length of directly extracted targets is difficultly controlled with length-varied overhangs, which may hinder the hybridization efficiency.

  1. Please make sure the abbreviation is explained when it is first introduced. For example, on page 2, pDNA is mentioned, but it is not explained what this means.

Response: It had been mentioned on page 2, “Experimentally, specific sequences used as probe DNA (pDNA)…”

  1. Please double check and correct, if needed, that an SH-group is required at the 5’-end of the 20-mer target DNA (page 3, section 2.2).

Response: Thank you for the reminder. That is a typing error. We have deleted it on page 3.

  1. In the equation mentioned in Section 2.4, the carboxyl group is shown as protonated, which would be minimal at pH 11.

Response: It is revised as below,

The reductive equation is represented as follows:

AuS(CH2)nX + e→ Au(0)+ S(CH2)nX, where X = COO and OH.

  1. In the legend to Figure 5, the sensor is referred to as “aptasensor”. Is it based on the aptamer sequence?

Response: Thank you for the correction. In Fig. 4 and 5, aptasensors is revised to genosensors.

  1. When the authors studied the time-dependence of the signal triggered by RNA extracted from 10^4 PFU/mL, the best reproducibility was observed for 20 min, and this time point was selected for subsequent experiments. Why it was only for 20 min that the data was so consistent? Why not for other time points?

Response: It is assumed the reason that the stability and reproducibility of hybridization is determined by the bonding strength and less non-specific adsorption. After the hybridization, the genosensors need to be rinsed to remove the non-specific adsorption. In shorter time the bonding strength of double strain structure is not stable enough, and in longer time the non-specific adsorption may increase, especially for full-length RNA fragments.  

  1. Could you please cite the source of a claim that “Generally, 1 PFU of DENV is equivalent to 600−1000 copies of virus particles”? I believed that 1 PFU corresponds to 1 viral particle, like 1 CFU corresponds to 1 bacterial cell, for example. Is it specific for DENV or for any virus?

Response: The definition of PFU is different from CFU. The ratio of 1 PFU to virus particles depends on the virus strains, culture conditions, and the length of time that the virus was stored. In the co-author laboratory, Prof. Guey-Chuen Perng, the DENV RNA copy numbers were about 600-1000 per 1 PFU. The article published in Journal of Virological Methods 86 (2000) 1–11 was found that each PFU for dengue virus should represent at least 100 or greater genomic equivalences [36], and the article published in Journal of Virological Methods 86 (2000) 1–11 mentioned that flavivirus RNA copy numbers were about 1000-3000 per 1 PFU [37].

  Moreover, the description is modified as “Generally, 1 PFU of DENV is equivalent to 600-1000 copies of virus particles in Dr. Perng’s laboratory, which depends on the virus strains, culture conditions and the stored time of virus [36,37]” on page 11..  

[36] H.-S.H. Houng, D. Hritz, N. Kanesa-thasan, Quantitative detection of dengue 2 virus using fluorogenic RT-PCR based on 3%-noncoding sequence, J. Virol. Methods 86 (2000) 1–11

[37] D.-Y. Chao, B. S. Davis, G.-J.J. Chang, Development of multiplex real-time reverse transcriptase PCR assays for detecting eight medically important flaviviruses in mosquitoes, J. Clin. Microbiol. 45 (2007) 584–589.

  1. Please explain how deltaRret ratio was calculated. What was the value for no-target control? If not zero, present the data for no-target control in Figure 8. Without listing the no-target control signal, it cannot be claimed “superior specificity”. Superior specificity means that all other serotypes would trigger the signal at the no-target level.

Response: The calculation of ΔRet and ΔRet ratio has been mentioned on page 8, “the Ret increment (ΔRet= Ret-tDNA-Ret-pDNA) and the ΔRet ratio (= ΔRet/Ret-pDNA) of genosensors after the hybridization of 1 pM and 1 nM tDNA are compared in Table 2”. It is also mentioned in Table 2 caption. ΔRet is defined as the Ret increment before and after hybridization. Here, Ret-pDNA is the no-target control and is used to normalize the ΔRet. The Nyquist plots of pre-hybridized genosensors have shown in Fig. 4-7 to obtain the Ret-pDNA values. Moreover, different pDNA/MHA/MCH-modified genosensors have different Ret-pDNA values. Therefore, the ΔRet ratio is used to compare the effect of MHA:MCH ratio on the sensing properties, shown in Table 2. The related description has been mentioned on page 9, “Compared to the ΔRet, ΔRet ratio is more often used to compare the ΔRet increment between different modified electrodes by normalizing the initial impedimetric status of pre-hybridized genosensors [29,30].”

The ΔRet ratio shown in Fig. 8 also uses the same calculation method via the blank measurement to obtain Ret-pDNA. Furthermore, the results obtained from the mixture of serotypes is added in Fig. 8. The description is revised as “The selectivity test of genosensors is shown in Fig. 8. The hybridization efficiency of the extracted RNA samples of each DENV serotype were respectively measured by three individual genosensors. The ΔRet ratio (56.4 ± 7.9%) for DENV1 is significantly larger than that for DENV2 (11.4 ± 3.4%), DENV3 (17.4 ± 2.3%) and DENV4 (14.7 ± 2.9%) (p<0.05 by student t-test). Moreover, the ΔRet ratio value for the mixture of DENV1+2, DENV1+3 and DENV1+4 was 58.4 ± 4.4%, 60.3 ± 5.0% and 57.8 ± 2.0% respectively. The calculated selectivity (=DENV1/DENV mixture) was in the range of 93.5% to 97.6%, indicating the good selectivity of the self-designed pDNA-immobilized genosensors to DENV1.”

  1. How in practice to interpret the results of the sensor? For example, if the signal is ~20%, how to differentiate between 10^2 PFU/mL DENV1 and 10^4 PFU/mL of DENV3?

Response: Statistic analysis, student t-test, can be used to identify the significance between two groups. In this case, the ΔRet ratio obtained from the 10^2 PFU/mL DENV1 and 10^4 PFU/mL DENV3 is respectively 25.7 ± 0.7% and 17.4 ± 2.3%, which has significant difference. The result suggests that the genosensors can distinguish the signal resulting from 10^2 PFU/mL DENV1 and have less interference from DENV3 concentration as high as 10^4 PFU/mL.

  1. How the sensor would respond to a sample with a mixture of serotypes (e.g. DENV1 and DENV4, or DENV2 and DENV3)?

Response: The results obtained from the mixture of serotypes is added in Fig. 8. The description is revised as “The selectivity test of genosensors is shown in Fig. 8. The hybridization efficiency of the extracted RNA samples of each DENV serotype were respectively measured by three individual genosensors. The ΔRet ratio (56.4 ± 7.9%) for DENV1 is significantly larger than that for DENV2 (11.4 ± 3.4%), DENV3 (17.4 ± 2.3%) and DENV4 (14.7 ± 2.9%) (p<0.05 by student t-test). Moreover, the ΔRet ratio value for the mixture of DENV1+2, DENV1+3 and DENV1+4 was 58.4 ± 4.4%, 60.3 ± 5.0% and 57.8 ± 2.0% respectively. The calculated selectivity (=DENV1/DENV mixture) was in the range of 93.5% to 97.6%, indicating the good selectivity of the self-designed pDNA-immobilized genosensors to DENV1.”

Reviewer 2 Report

The paper reports on a DNA impedimetric sensor for the detection of extracted RNA Dengue virus. The paper could be of interest for Sensors' readers but it requires major revisions.

Gold electrodes functionalization by MHA/MCH and RNA/DNA sensing are not new, so that the novelty of the paper is not a strong issue of this work. I suggest to add a comparison between the existing similar results in literature and those presented by authors, adding a table and commenting pros and cons.

Errors must be written everywhere in the text in the proper way. Errors must have only one significant digit (at least two, if the first is 1). The numbers 5578.00 ± 163.57 kΩ are not meaningful and they must correctly written as 5580 ± 160 kΩ. Please, correct all errors accordingly. 

Figure 3, 4, 5, 6, 7 and 8 are partially overlapped to the text and they cannot be read properly. Please, check the pdf of the paper before re-submitting it. 

Author Response

The paper reports on a DNA impedimetric sensor for the detection of extracted RNA Dengue virus. The paper could be of interest for Sensors' readers but it requires major revisions.

Comments:

  1. Gold electrodes functionalization by MHA/MCH and RNA/DNA sensing are not new, so that the novelty of the paper is not a strong issue of this work. I suggest to add a comparison between the existing similar results in literature and those presented by authors, adding a table and commenting pros and cons.

Response: Thank you for the comment. The description is added on page 12&13, Table 3 compares this study with previous electrochemical genosensors for the DENV detection. Most DENV genosensors detected synthesize DNA [15,18-20] and RNA amplicon [26]. Except for Jin’s study [16], there are few studies to directly detect the extracted RNA without the process of nucleic acid amplification. Moreover, the pDNA/MHA(0.04 mM)/MCH(1 mM)/Au genosensors have better sensing properties than Jin’s sensor.

Table 3. Comparison of the sensing properties of different electrochemical genosensors for the detection of DENV.

Electrode

Method/Target

Dynamic range

LOD

Ref.

NH2-pDNA/APS/Pt/AAO

EIS or DPV/synthetic 30mtDNA

1 pM–1 μM

2.7 pM

[15]

pDNA/PGE

DPV/synthetic 22mtDNA

10–100 nM

3.1 nM

[18]

NH2-pDNA/chitosan/ZnO /Pt-Pd/FTO

CV/synthetic 35mtDNA

1–100 μM

43 μM

[19]

NH2-pDNA/MPA/Mn2O3 nanofiber/GCE

EIS or DPV/synthetic consensus dengue primer

1 aM–1 μM

0.12 aM

[20]

ITO

LSV/pDNA-RNA amplicon mixture

--

2 amol

[26]

pDNA/SiO2@APTES-GO/Pt

EIS/extracted RNA

--

1 fM

[16]

NH2-pDNA/MHA/ MCH/Au

EIS/extracted RNA

102–105 PFU/mL

20 PFU/mL (~20 aM)

This work

AAO: anodic aluminum oxide, APS: 3-aminopropyltrimethoxysilane, DPV: differential pulse voltammetry, FTO: fluorine doped tin oxide, LSV: linear sweep voltammetry, MPA: mercaptopropionic acid, PGE: pencil graphite electrode

  1. Errors must be written everywhere in the text in the proper way. Errors must have only one significant digit (at least two, if the first is 1). The numbers 5578.00 ± 163.57 kΩ are not meaningful and they must correctly written as 5580 ± 160 kΩ. Please, correct all errors accordingly.

Response: Thank you for the comment. We have corrected the writing of errors.

3.Figure 3, 4, 5, 6, 7 and 8 are partially overlapped to the text and they cannot be read properly. Please, check the pdf of the paper before re-submitting it.

Response: Thank you for the reminder. All the figures and legends have been adequately adjusted.

Round 2

Reviewer 1 Report

The authors addressed most of the questions raised based on the original manuscript submission. Nevertheless, some questions remain. For example, the authors explained that the signal is expressed in ΔRet ratio, which is calculated as ΔRet/Ret-pDNA. But in the y-axis of the graphs in Figures 6b, 7b,8 is expressed in ΔRet ratio (%). How the value was converted to %? What value was taken as 100%? Please add the description to the manuscript.

Author Response

The authors addressed most of the questions raised based on the original manuscript submission. Nevertheless, some questions remain. For example, the authors explained that the signal is expressed in ΔRet ratio, which is calculated as ΔRet/Ret-pDNA. But in the y-axis of the graphs in Figures 6b, 7b,8 is expressed in ΔRet ratio (%). How the value was converted to %? What value was taken as 100%? Please add the description to the manuscript.

Response: Thanks for the reminder. The calculation is ΔRet ratio (%)= ((Ret-tDNA-Ret-pDNA)/Ret-pDNA)´100). We revise the definition of ΔRet ratio (%) on page 8, “…the Ret increment (ΔRet= Ret-tDNA-Ret-pDNA) and the ΔRet ratio (%)(= (ΔRet/Ret-pDNA)´100) of genosensors” and on page 10 Table 2. Because the Ret-tDNA value is larger than the Ret-pDNA value, the Ret increment is a positive percentage value. If the ΔRet ratio (%) is 100%, the Ret-tDNA value is twice the Ret-pDNA value.

Reviewer 2 Report

In the revised form, the paper can be accepted for publication.

Author Response

In the revised form, the paper can be accepted for publication.

Response: Thank you for the decision.